# Pediatric Hematology and Oncology Center Integrated by Telemedicine: Experience, Challenges and First Results of a Cross Border Network

**DOI:** 10.3390/healthcare11101431

**Published:** 2023-05-15

**Authors:** Tabea Troschke, Aleksandra Wieczorek, Konrad Kulinski, Tomasz Ociepa, Karolina Zielezinska, Holger N. Lode, Tomasz Urasinski

**Affiliations:** 1Department of Pediatric Hematology and Oncology, University Medicine of Greifswald, Ferdinand-Sauerbruch-Strasse, 17475 Greifswald, Germany; 2Department of Pediatric Oncology and Hematology, Jagiellonian University Medical College, 265 Wielicka Str., 30-663 Krakow, Poland; a.wieczorek@uj.edu.pl (A.W.); konrad.kulinski@uj.edu.pl (K.K.); 3Department of Pediatrics, Hemato-Oncology and Gastroenterology, Pomeranian Medical University, Ul. Unii Lubelskiej 1, 71-252 Szczecin, Poland; tociepa@gmail.com (T.O.);

**Keywords:** telemedicine, pediatric oncology and hematology, cross border education, Interreg, funding, binational medicine

## Abstract

This article reports on the development, implementation and management of a German–Polish telemedicine network in the field of pediatric oncology and hematology in the Euroregion Pomerania. The achievements and challenges of joint medical case reviews involving patients and their care givers, as well as cross-border education activities for physicians, students and nursing staff, are presented. In addition to a progress report, the results of an evaluation of the participants and teachers, likewise the measurement of knowledge growth, are given.

## 1. Introduction

Pomerania is a region on the shore of the Baltic Sea with an eastern German and a western Polish area, and that coined the term “Euroregion Pomerania” for a co-operative entity between Germany and Poland since 1995. The purpose of strengthening this region is to promote common interests, supported by funds from the Interreg programs promoting cross-border cooperation in the European Union. One particular characteristic of the Pomerania region is its scarce population as it ranges amongst the lowest population densities in Germany and Poland (Mecklenburg–Western Pomerania: 69 persons/km^2^, West Pomerania: 72 persons/km^2^, Lesser Poland: 226 persons/km^2^) which is associated with the risk of medical undersupply and the need for cross-border networking and telemedicine [1,2].

Therefore, the Euroregion Pomerania has great experience in telemedicine, with a unique cross-border telemedical network concerning different areas of medicine, established in 2001 and joining different specialists. A German-Polish telemedicine network in Pomerania was realized with a focus on videoconferencing using high-speed internet, which allowed for the simultaneous transport of X-ray studies, pathology slides, endoscopy images and documents. This network included numerous larger medical areas, but so far there has not yet been a structured cross-border telemedical care concept in the field of pediatric oncology and hematology [3,4,5].

To improve the quality of care for patients with these extremely rare disorders the pediatric OncoGroup was formed at the end of the last millennium, which is the network of pediatric oncologists and hematologists in Pomerania aiming to achieve this goal through discussions about complicated and critical cases. The incidence of pediatric cancer is rare (with 1200 new patients diagnosed every year in Poland and 1800 in Germany), and it encompasses over 200 different disease entities, each of which fulfills the criteria of orphan disease and requires highly specialized diagnostic and therapeutic procedures. For example, neuroblastoma, which was one of the main of points of interest of the project, accounts for about 6–8% of all childhood cancers, with about 180–200 patients diagnosed every year in both countries. In the patients from the high-risk group, the intensive multimodal therapy is required, including autologous stem cell transplantation and immunotherapy, and in some cases also therapy with 131-I meta-iodobenzyl-guanidine (MIBG) [6,7,8].

The frequency in the need to use all these methods is relatively low, but all of them need high experience which may be difficult to gain with a comparatively low number of patients in the single medical centers of the Euroregion and this situation forced specialists to interact even more closely and share knowledge in order to provide optimal care to children with cancer. The need for a cross-border telemedicine network for children with cancer was therefore obvious and further advancements of a telemedicine network in the areas of specialized medical interactions, education, as well as the inclusion of patients, were considered important assets to address these challenges. The given developments fostered the project proposal to build a pediatric hematology and oncology center in Pomerania integrated by telemedicine with the aim to further advance the quality of care for children with the cancer in the region. The aim was also that the affected families should not incur any further costs as a result of the improvement in care. To develop a structure for a telemedicine network, a cooperation between the Medical University in Greifswald, the Pomeranian University in Szczecin and the Jagiellonian University Medical College in Krakow was established and the Temicare project was funded in the Interreg V A Cooperation Program Mecklenburg-Vorpommern/Brandenburg/Poland from 2017 onwards.

As Beuken et al. showed, patient involvement in the decision-making process regarding their healthcare, communication with the healthcare providers and information throughout the healthcare process are crucial for successful cross-border healthcare. Therefore, the aim of the project was to establish a teleinformatic system which enables real-time online patient consultations in the connected centers in Poland and Germany, with the possibility for patients’ families to participate in such consultations, and also to allow for “bed-site” consultations and sharing of imaging, with efforts to secure data privacy according to the European law. The subject of consultation was extended to involve patients with all cancers as well as hematological diseases. The additional aim was to establish the German–Polish pediatric oncology center, in which doctors and nurses from both countries worked in a rotational system, and the possibility for the education of students, physicians and medical personnel. The main aim of this education was to increase knowledge concerning mainly immunotherapy and radionuclide therapy. The Temicare project was established to provide a platform to improve the quality of care for children with cancer treated on the both sides of the Polish–German border. A special focus was placed on neuroblastoma, since the partners have clinical and research expertise about this challenging childhood malignancy. However, the forum for medical exchange and education was explicitly open to all topics of pediatric hematology and oncology and the output of clinical discussion and educational topics reflect the broad spectrum of the field [9,10,11,12].

## 2. Materials and Methods

### 2.1. Preparation of the Temicare Project

The active search for partners for a telemedically integrated German–Polish network began in 2017. Linked to the OncoGroup network of pediatric oncologists and hematologists, a close exchange took place about the demands and design possibilities of an organized structure. While some institutions decided against a more permanent partnership (among other things due to the high organizational effort), a group of the following institutions finally manifested itself:The Leader—Department of Pediatric Hematology and Oncology, University of Medicine in Greifswald, Germany (Greifswald), with great experience in treatment of pediatric oncology patients, with a special focus in neuroblastoma immunotherapy, and partners.Department of Pediatrics, Hemato-Oncology and Gastroenterology, Pomeranian Medical University, Szczecin, Poland (Szczecin I), as the biggest pediatric oncology and hematology department in the region of Western Pomerania on the Polish side of the border, treating children with cancer from the whole region on the Polish site. *Department of Pediatrics, Oncology and Immunology, Pomeranian Medical University, Szczecin, Poland (Szczecin II). *Department of Pediatric Oncology and Hematology, Jagiellonian University Medical College in Kraków, Poland (Kraków), invited to the project as an external partner as the Polish national coordinating center for neuroblastoma therapy, with long-time established cooperation with the Greifswald University of Medicine.


** Until 2017, there was only one pediatric hematology and oncology center in the Western Pomerania region: the Department of Pediatrics, Hematology and Oncology. In 2017, as a result of the restructuring of the university and the hospital, two new units were established: the Department of Pediatrics, Hemato-Oncology and the Gastroenterology and Department of Pediatrics, Oncology and Immunology. Teams in these two units were created from the experienced doctors and nurses who previously worked in the “mother” department. Both units are located in the same building and work closely together taking care for children with malignant and non-malignant hematopoietic system disorders (Dept. of Hemato-Oncology) and with solid tumors (Dept. of Oncology).*


The group of participating institutions thus finally included the locations Greifswald, Szczecin and Krakow (Figure 1).

The first meeting was held in Greifswald in October 2018 and it was clearly stated by members of all participating centers at the early stage of the project preparation that not only a trusting, integrative cooperation with families, but also education in the field of pediatric oncology and hematology, will remain pivotal elements of the project. In the vivid discussion during several online meetings in the preparation phase, the most important issues were defined by the members, including the aim of the project, the target groups, the forms of online education, the curricula and the timetables of all planed educational events.

Aims of the project:To involve families in the decision-making process of diagnosis and therapy;To create, develop and introduce the online hybrid system of additional education in the field of pediatric oncology and hematology for all centers participating in the project with the use of the internet network connecting all centers;To assess the feasibility and efficacy of this form of education with the use of the tools created for this specific purpose.

Target Groups:Parents and care givers of patients;Staff physicians;Residents in pediatrics as well as in pediatric hematology and oncology;Students in training (regular and practical) in pediatrics;Members of the OncoGroup (the abovementioned pediatric oncologists and hematologist from Greifswald, Rostock, Schwerin and Szczecin);Staff nurses from the centers participating in the project.

Use of language:

Different communication options were planned to be used during the meetings: English as the common language for partner meetings and case discussions and both Polish and German with the presence of interpreters when more participants would be present, especially patients’ families.

After the approval of the funding by the Interreg-VA program, the actual activity of the cooperation under the name of “Telemedicine integrated, German-Polish Children’s Cancer Center in the Euroregion Pomerania; Temicare”; (ger. Telemedizinisch integriertes, deutsch-polnisches Kinderkrebszentrum in der Euroregion Pomerania; pol. Telemedycznie Zintegrowane, Polsko-Niemieckie Centrum Onkologii i Hematologii Dziecięcej w Euroregionie Pomeranii)” began on 1 January 2020.

As digitization is a prerequisite of telemedicine in order to provide medical services over a distance, one of the first steps was to develop digitization in all centers using the funds of Interreg VA program. The infrastructure was agreed upon and tested by the local IT services to ensure good cooperation and quality of connection. The type of IT infrastructure, equipment for storage and network connections were also discussed and agreed. Affected by the obstacles of the pandemic, the telemedical linking of the partner institutions was realized, in particular, via the procurement and installation of video conference systems, software and other telemedical devices. The whole structure was also taking into account the country-specific data protection regulations. For this purpose, a legal opinion was prepared by a specialist law company as part of the project.

At the same time, the working groups met at regular intervals of 3–4 weeks to develop target-group-specific curricula. A teaching canon for physicians, students and nurses was developed by asking representants of the institutions about their needs. The curricula were developed up jointly by specialist representatives from all facilities (such as senior physicians and ward managers) and included, on the one hand, topics in which individual centers specialize and with which other facilities have only dealt with peripherally within the training so far, but also fundamental topics in pediatric oncology and hematology, which all participants should know in order to align with the treatment standards across borders.

It has been also decided that, except of the period of the strict COVID-19 pandemic lockdown when the students’ live activities were suspended, the education will have a hybrid form: participants present personally in the center will observe and communicate with the speaker and other participants using the online network. Special facilities, e.g., seminar rooms supplied with high resolution interactive screens, were created in all participating centers using funds from this project.

As soon as the video conferencing technology was set up and tested and the curricula were developed, the implementation of the planned telemedical activities could begin.

### 2.2. Implementation of the Cross-Border Network

The central aspects of the cooperation were the telemedical case discussions as well as the joint education.

#### 2.2.1. Joint Medical Case Presentations with Family Integration

The inclusion of patients and their care givers is an essential component in pediatric care. Patient families on the wards were informed by their doctors about the planned telemedical cross-border meetings related to their child and offered participation in the discussions. Before the meeting, an informed consent procedure was a prerequisite to the participation of the families. The format of the meetings included a detailed presentation about the history of the disease as well key questions to the expert panel invited to the meeting. These questions were then systematically addressed by the experts and main conclusions and action points were identified related to the case. This was then followed by questions asked by the families and answered by the experts. In order to educate in the area of patient communication, students were also invited to the case presentations. These joint medical case presentations with family involvement were also part of the educational offer.

#### 2.2.2. Creation of Courses for Physicians, Nurses and Students

Gradually, the various activities were established within the framework of the cooperation. Forms of hybrid education included:Seminars;Case presentations;Telemedicine online OncoGroup meetings as a continuation of the activities of the OncoGroup.

Curricula as the basis of the seminars

It was accepted by all participants of the project that seminars for physicians and medical students should follow the formula of “The recent advances in …”. and/or “How do I treat …”, bringing the attendees the “up-to-date” knowledge in pediatric oncology and hematology and allowing the presenters to share their personal experience and expertise. It was believed that this form would attract participants making the seminars’ content more interesting. Topics of seminars for nurses were selected in a more practical manner in order to discuss similarities and differences of routine procedures performed by nurses on pediatric oncology and hematology wards. The list of seminar topics for students, physicians and nurses is presented in Table 1, Table 2 and Table 3.

Case presentations as a part of education

As mentioned above the medical case discussions were also part of the medical education offer. Therefore, the case selection for discussion was left to the decision of participating centers with the assumption that it will allow the presentation of the most representative cases from an educational point of view, but also of those that were unusual and challenging. The formula of questions: “What would you do?” was preferable. Authors of case presentations were encouraged to include anonymized original imaging studies, histopathological and hematological slides as well as lab results instead of their descriptions to stimulate discussion. It was stated that case presentation should be prepared with respect to personal data protection regulations.

Organizational aspects

It was decided that case presentations should not last longer than 25 min (15 min for the presentation plus 10 min for the discussion), and seminars no longer than 60 min (45 min for the presenter plus 15 min for the discussion). The telemedical online meetings were designed to last about three hours. In addition, internship opportunities were created for the target groups at the partner institutions. In order to improve the educational potential of the project some of events were run by junior physicians supervised by specialists.

## 3. Results

### 3.1. General Results

The telemedical linking of the partner institutions was successfully completed. As part of the network, 90 joint events with a total of 2295 participations were held between 1 January 2020 and 30 June 2022. These data are presented in Table 4.

### 3.2. Integration of Families into the Network Activities

Pediatric care, in particular the care of children with cancer, largely involves detailed, extensive and time-consuming communication with care givers (mainly parents). It is a key component in building a relationship between doctors and patients and their family. Therefore, the aim was to integrate care givers in the decision-making process within the telemedicine network activities as described in the methods section. During the telemedicine meetings, options for further treatment were suggested and the associated opportunities and risks were presented, and the families were given space for detailed queries. In dialogue between the treating doctor on site and the families, these arguments could be included in their joint decision on the next steps.

The realization of these approaches showed a high organizational effort in advance of finding dates to prepare materials in Polish/German associated with a limitation in the number of meetings involving parents. The first impressions indicate that participating families were highly satisfied with the procedure and the content of the meetings as well as with the opportunity to witness the expert panel’s work and to ask questions. In those cases where the families were not present in the meeting, the specialist presenting the case provided feedback about results of the discussion/consultation for the families.

In December 2023, five families were asked to provide a written assessment of the case discussions using a short questionnaire. All families indicated that it was very important for them to hear the opinion of other doctors on their child’s medical case. One parent emphasized that the meeting cleared his doubts about the further treatment of his child. Furthermore, the possibility of verification and obtaining more detailed background information on studies of drug use was formulated.

### 3.3. Joint Education Offers Using the Examples of Case Presentations and Seminars for Students and Physicians

#### 3.3.1. Assessment and Assessment Tools

In order to measure the success and potential for improvement in the joint teaching activities, the change in the level of knowledge was determined by pre-seminar and post-seminar tests in the seminar offer. In addition, an evaluation was carried out for both the participants and the teachers for the case presentations (CP) and the seminars offered for students and doctors (S).

The following two methods of assessment have been selected to assess the efficacy of the forms of education presented above:(1)All presenters of the seminars were obliged to prepare a test with three multiple-choice questions related to the topic. Participants were asked to solve the test before the seminar and repeat it when the seminar was over. Answers were compared and analyzed. Tests scoring 0 or 1 correct answers were considered negative, whereas those scoring 2 or 3 were classified as positive.(2)To assess the participants’ contentment, the attendees were asked to scan a QR code that connected them with an online survey to answer several questions designed to measure their opinion regarding the content and form of the presentation. In addition, participation was possible via a paper version of the questionnaire. These data were subsequently analyzed by the project coordinator.

All tests as well as contentment assessments were anonymous.

#### 3.3.2. Results of the Pre-Seminar and After-Seminar Tests

During 14 meetings between 5 November 2021 and 30 June 2022, altogether 354 pre-tests and 357 after-tests were filled out. The results of the pre-seminar and after-seminar tests are presented in Table 5. An analysis of pre-seminar tests revealed that the rates of participants who received negative results ranged from 0 to 0.91 (mean 0.41; median 0.41). Only 1/14 pre-seminar tests were passed by all participants.

The results of the post-seminar tests were much better. The rate of those who passed it (score 2 or 3) ranged from 0.31 to 1.0 (mean 0.87; median 0.92). The participants of all seminars passed post-seminar tests, while 4/14 tests were passed by all (100%) of participants. The rate of those who transferred from the group of negative to positive responders ranged from 0.143 to 0.80 (mean 0.2712; median 0.286). In the case of 2/14 seminars, the rates of those who passed post-seminar test were smaller by 2.4 and 14.3% than those who passed pre-seminar test. Using Student’s *t*-test for paired samples, it was shown that the mean rates of those who received positive scores (2–3) were statistically higher in post-seminar tests as compared to pre-seminar results (0.59 vs. 0.87; *p* < 0.001). These data are illustrated in Figure 2.

The results of the pre-tests show that the selection of the topics to be discussed was in line with the expectations of most of the listeners.

#### 3.3.3. Assessment of the Project Participants’ and Lecturers’ Contentment

Between 28 March and 1 April 2022, 6 surveys of participant contentment (lecturers and participants separately) of the Temicare hybrid meetings were conducted, 3 after the case presentations (CP) and 3 after the seminars (S). A total of 147 participants were surveyed (87 participants of CP and 60 participants of the S). In addition, in 15 cases, the lecturers were surveyed (7 after CP and 8 after S). While the evaluation of the participants was limited to the survey period, the evaluation of the lecturers referred to all seminar offers for students and doctors carried out since 5 November 2022.

In total, 68% of CP participants and 87% of S declared that the level of their knowledge improved and that they gained some new information. A total of 57% of CP participants and 87% of S stated that they acquired new strategies that were useful in their practical work. Similar rates (54% of CP and 81% of S) recognized that Temicare events helped them to feel better prepared for their future practical activities.

Moreover, 100% percent of lecturers admitted that, due to participation in these meetings, they deepened their knowledge and, having had the chance to exchange their expertise with colleges from other centres, gained new inspiration.

Language was not a barrier for most participants, because 96% of CP and 100% of S participants acknowledged that they were able to follow CP and S run in English. This was confirmed by lecturers who assessed that the knowledge of English among students is at least good or even very good. However, 65% of CP and 87% of S student participants declared that they would be willing to participate in a course for medical English. These rates were much smaller among lectures ranging from 29% for CP to 25% for S presenters, respectively.

### 3.4. Identified Barriers

The high coordinative and time expenditure that arose in connection with all project activities could be identified as one of the main barriers in the cooperation. Due to different work and training structures at the partner institutions, a high degree of constant exchange and willingness to compromise was necessary. The intensive discussion regarding the requirements of data protection and the institute’s internal requirements for the use of technology took place within the framework of a specially formed working group, “Telemedicine and data protection”, and was significantly facilitated by the preparation of a legal opinion by a specialist law firm.

The preparation (particularly for the bilingually implemented joint activities) required extensive coordination and a long lead time. In the implementation of joint courses, it proved challenging to choose topics and time frames that were relevant and feasible for everyone involved within the existing teaching structures. The coordination of the data protection requirements and limits, as well as the creation of bilingual information and consent documents, required extensive preparations.

At the same time, the pandemic situation demanded flexibility and limited the planning of personal visits to certain times of the year. Language barriers in personal exchanges could be overcome with the help of interpreters.

An overview of the activities and the associated barriers is presented in Table 6.

## 4. Discussion

There is growing evidence that telemedicine services play a significant role in improving the quality of professional healthcare for both children and adolescents with cancer. There are reports on binational cooperation, such as the long lasting one between Iraq and Italy presented recently by Al-Jadiry et al. The authors summarized their experience, stating that this form of exchange of knowledge and expertise across continental boundaries meaningfully improved the diagnoses and management of pediatric cancer at the Children’s Welfare Teaching Hospital in Baghdad. There are also global networks such as the Global Neuroblastoma Network; the results of its activity show the utility of an international online tumor board focused on challenging pediatric tumors, especially for countries with low and middle incomes where local expertise may be limited. The systematic review by Shah A.C. et al. revealed 32 telemedicine studies on malignant and non-malignant hematology; however, only 7 of them addressed the pediatric population. The authors of this review concluded that more research is needed to determine the efficacy of implementing the more widespread use of telemedicine for hematological conditions. We believe that the results presented in this manuscript add some more arguments speaking in favor of the utility of telemedicine in the field of pediatric oncology and hematology [13,14,15].

The results of the German–Polish cooperation so far show that the binational telemedical network includes both the patients and their families as well as the medical staff involved. The phase of the technical and organizational network setup proved to be very time-consuming and communication-intensive. Appropriate measures could be developed to overcome further barriers (such as language, different structures, requirements for data protection), so that a general transferability of such a cooperation to other hospitals can be assumed.

With regard to the involvement of families in the medical processes, the first impressions suggest very positive effects. In particular, a high degree of transparency for those affected can be assumed in this way, combined with a strengthening of the doctor–patient relationship. However, this offer, together with the comprehensive preparation and follow-up processes, must be established more comprehensively in order to finally be able to meaningfully check this initial feedback from the families within the framework of a somewhat larger group of participants.

The success of German–Polish teaching in the area of seminars for students and doctors could be proven. Both the results of the knowledge tests, and the surveys of those involved, showed that cross-border teaching in the field of pediatric oncology and hematology is not only possible, but can also be profitable for all sides. In further investigations, analyses would be necessary of whether such results can also be verified for other courses and target groups (such as nursing staff). As part of the study, a direct increase in knowledge was determined. However, in order to make the long-term effects measurable, a follow-up study to check the durability of the increase in knowledge would be necessary.

It can be assumed that, due to the professional exchange between the institutions involved, patients in the field of pediatric oncology and hematology care can not only be covered, but also improved where it already existed. Savings can therefore also be expected on the part of the service providers by avoiding under-, over- or incorrect supply. Against the background of the advantages of the cooperation that have been identified so far, it seems advisable to expand the telemedicine network with other partner institutions in order to expand the catchment area for patients and medical staff who benefit from the activities and, at the same time, to expand the network using the know-how of other medical institutions.

## 5. Conclusions

Looking from the perspectives of telemedicine development, this pandemic showed clearly how important it is to have established network connections between the centers, with close cooperation, compatible devices and further development. The lack of possibility of travelling and meeting in person did not influence the regular patients’ consultations and meetings necessary for project coordination.

The telemedicine concept proved to be effective for all aims of the project. It provided the facilities for regular partner meetings, the possibility for discussions concerning currently treated patients, and the opportunity for patients’ families to join meetings online and discuss treatment options with all partners. Telemedicine was also a good option for physicians’ and nurse’ education, as it allowed for the meeting of a relatively high number of staff members, without the necessity for travelling. Saving the time needed for travelling is always a very desirable option in the situation of low numbers of staff, which is a common problem in all pediatric oncology departments.

The Temicare project clearly showed that the use of appropriate IT tools combined with expert knowledge in the field of pediatric oncology translates into real results and plays an increasingly important role in the development of science and the educational process, and its further development is indisputable and inevitable.

From the patients’ point of view, too, telemedical networking with other specialist facilities, including a translation service, represents an innovative opportunity to benefit binationally from the skills of other specialists and thus experience more security and trust in the treatment process. Even in cases of rare oncological and hematological diseases, local care can be improved and additional hurdles (such as traveling to obtain a second opinion) can be avoided. It is therefore advisable to expand the circle of families who benefit from this opportunity by expanding the Temicare network in order to optimize patient care in the field of pediatric oncology and hematology across borders. The Temicare project can thus serve as a good practice example for creating similar networks in other parts of the world with similar challenges, such as Scandinavia or Latin America.

## Figures and Tables

**Figure 1 healthcare-11-01431-f001:**
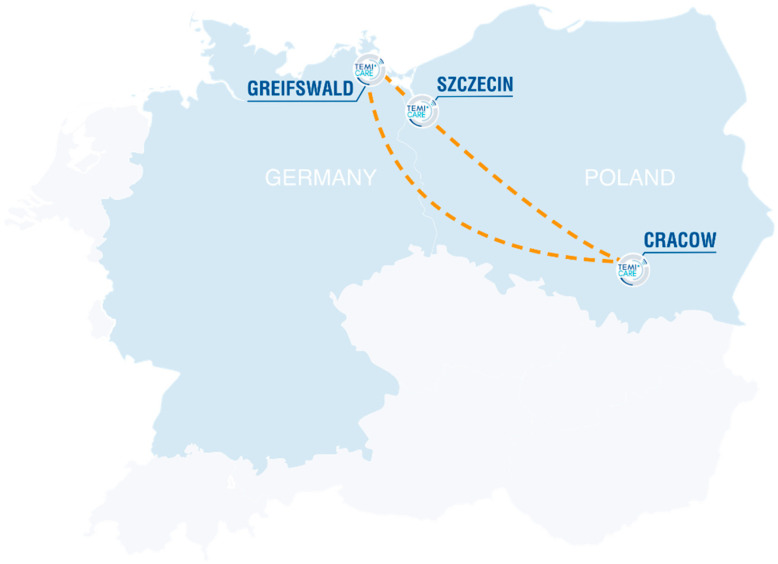
Locations of the telemedical network Temicare.

**Figure 2 healthcare-11-01431-f002:**
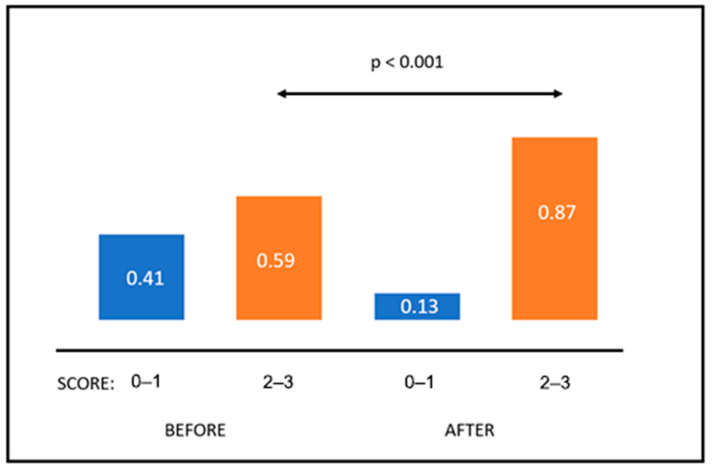
Mean rates of negative and positive scores in tests performed before and after the seminars.

**Table 1 healthcare-11-01431-t001:** Curriculum students.

Solid Tumors	Leukemia/Hematology	Oncological Emergencies, Complications, Infections
Childhood Lymphoma	Disorders of coagulation	Emergencies in pediatric oncology
Diffuse intrinsic pontine glioma in children	New treatment options in leukemia	Invasive fungal infections
Neuroblastoma 4s— diagnosis and management	Acute myeloblastic leukemia– new treatment approach AML in special groups (down syndrome), APL	Flow cytometry in leukemia– diagnosis
Hepatocarcinoma in children	Anaemia–differential diagnosis and therapy	Supportive treatment in pediatric oncology
Hodgkin lymphoma— new treatment options, relapse treatment, special groups	Causes of neutropenia in children	
When to consider a brain tumor	Immune thrombocytopenia	
Neuroblastoma— how to diagnose, how to treat	Approach to childhood leukemia	
	Immunotherapy in pediatric oncology	

**Table 2 healthcare-11-01431-t002:** Curriculum physicians.

Management of non-malignant hematological disorders in children. Diagnosis and treatment of central nervous system tumors in pediatric oncology. Solid tumors in childhood—differential diagnosis and current treatment strategies. Challenges with managing acute leukemia of children. Consensus forum on rare malignancies in pediatric oncology.

**Table 3 healthcare-11-01431-t003:** Curriculum nurses.

Handling of central venous catheters (Broviac and vascu-port)—blood samples collection, administration of drugs, dressing change. Protection against adverse effects of cytotoxic drugs. Nurses procedures before, during and after transfusion of blood products. Diagnostic and therapeutic employment of iodine-123 and 131-MIBG. Enteral and parenteral nutrition, diets in pediatric oncology. Patients care after blood stem transplantation. Nurse care for patients needing isolation.

**Table 4 healthcare-11-01431-t004:** Numbers of hybrid online events held within the frames of the telemedicine project and numbers of participants.

Type of Event	Number of Events	Number of Participants	Participants/Event (Average)
Seminars for students/physicians	14	565	40.2
Seminars for nurses	5	77	15.4
Case presentations	58	1365	23.5
Telemedicine Online Meetings	8	119	14.8
Conferences	2	102	51
Internships	3	7	2.3

**Table 5 healthcare-11-01431-t005:** Rates of negative and positive pre- and after-telemedicine seminar tests results.

	BEFORE-SEMINAR TEST		AFTER-SEMINAR TEST	
	NEGATIVE (Score 0–1)	POSITIVE (Score 2–3)	NEGATIVE (Score 0–1)	POSITIVE (Score 2–3)
SEMINAR				
1	0.40	0.60	0.10	0.90
2	0.65	0.35	0.08	0.92
3	0.44	0.60	0.00	1.00
4	0.00	1.00	0.10	0.86
5	0.26	0.74	0.16	0.84
6	0.06	0.94	0.08	0.92
7	0.61	0.40	0.28	0.72
8	0.31	0.69	0.00	1.00
9	0.43	0.57	0.00	1.00
10	0.09	0.91	0.00	1.00
11	0.91	0.10	0.69	0.31
12	0.55	0.45	0.28	0.72
13	0.22	0.78	0.07	0.93
14	0.80	0.20	0.00	1.00
MEAN	0.41	0.59	0.13	0.87
MIN.	0.00	0.10	0.00	0.31
MAX.	0.91	1.00	0.69	1.00
MEDIAN	0.41	0.60	0.26	0.92

**Table 6 healthcare-11-01431-t006:** Overview of project activities, goals and outcomes.

Telemedical Network cross Border Activity	Involved Group (s)	Goals	Main Output Indicator	Outcome	Facilitators	Barriers
General technical connection of all network institutions.	Physicians, students, nurses.	Immediate uncomplicated regular medical exchange. Savings in travel and time costs.	Establishment.	Established successfully.	Funding support for the telemedical equipment and staff by Interreg-VA.	Need for intensive coordination in the areas of IT and data protection.
Case discussions	Physicians.	Medical exchange with other experts.	Feasibility.	Established successfully. Positive feedback from the evaluations.	Physicians received professional support through exchanges with other experts.	Research and translation of materials. Compliance with data protection regulations.
	Patient and their care givers.	Participation in the process of diagnosis and treatment in terms of a better understanding. Avoiding travelling time and expenses.	Feasibility.	Feasible.	Overcoming language barriers with a translator.	Intensive preparations in advance (e.g., translation of materials).
	Students.	Insights into the professional activities.	Evaluation by feedback.	Established successfully. Positive feedback from the evaluations.	Good English knowledge.	Cross-institutional scheduling.
Seminars for students and doctors	Physicians, students.	Exchange of special knowledge for teaching purposes.	Evaluation by feedback.	Established successfully. Positive feedback from the evaluations.	Good English knowledge.	Cross-institutional scheduling.
Education offers for nurses	Nurses.	Imparting teaching content, harmonization of treatment standards.	Feasibility.	Established successfully.	Overcoming language barriers with a translator.	Cross-institutional scheduling. Limited staff availability due to the need for patient care. Long-term plannings necessary.
Telemedicine online meeting	Physicians from the Temicare network and other cooperating clinics in Germany and Poland.	Exchange of medical experiences.	Feasibility.	Established successfully, alternately in Germany and Poland.	Expansion of network activity.	-
Cloud with recorded seminar content	Physicians, students, nurses.	Flexible access to the teaching materials.	Feasibility.	Established successfully.	Flexibility in terms of time and place of learning.	Compliance with data protection regulations.
Internships in partner institutions	Physicians, students, medical staff.	Extension of professional experience.	Feasibility.	Performed successfully.	Increase in identification with the network through interpersonal exchange.	Pandemic limitations in travellings. Long-term plannings necessary. Language barriers.

## Data Availability

The data will be provided on request by the team of authors.

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
