# Peer review of "Pediatric Hematology and Oncology Center Integrated by Telemedicine: Experience, Challenges and First Results of a Cross Border Network"

_healthcare, 2023, doi:10.3390/healthcare11101431_

Round 1

Reviewer 1 Report

This project report discussed the experience, challenges, and results of the cross-border telemedicine network in Pomerania, a region next to eastern German and western Polish area. The report mentioned several barriers in the cooperation, I wonder from the perspective of patients, could the health insurance cover their treatment through telemedicine in neighboring county? Do they have high-speed internet to use telemedicine? What types of hematology and oncology could be used for telemedicine? These questions may need to be introduced in the Introduction section.

 In the Introduction section, second sentence, “the purpose” of what?

In the discussion section, line 359, “it would have to be analyzed” -> “it would have to analyze”? there are also some other grammar issue, please double check.

Author Response

Dear reviewer,

Thank you for your constructive advice and questions.

  • We have added information on costs (line 66/67) and
  • high-speed internet (line 38).
  • Line 80/81 describes that "all cancers as well as hematological diseases" can be part of the telemedicine consultations.
  • We have completed the sentence "The purpose of…",
  • also corrected the "Analyze" sentence (line 406) and
  • revised the grammar.

Best regards

Tabea Troschke

Reviewer 2 Report

Check the dates on line 278, are these correct?

It is no clear enough how was the impact of telemedicine in parents' decision making. Could you describe it?

In reference to the manuscript “Pediatric Hematology and Oncology Center Integrated by Tele medicine: Experience, Challenges and First Results of a Cross Border Network” that I previously reviewed, my comments are the following:

1.     This telemedicine border network between two countries with a low population density (but with challenging cases of children cancer) is a very necessary tool for sharing experiences, knowledge, comprehensive diagnosis, and to include parents and family in decision making.

2.     Use this network as an educational strategy and to implement a pediatric oncology center based in the networking between two countries is innovative. Systematically, cooperation through border.

3.     Discussion must be completed because there is no comparation with other telemedicine teams in the world.

4.     Conclusions must be focused on the benefit is oncological patients specific benefits rather than benefits of telemedicine (cost, distance, etc,…)

5.     I consider table 5 is not necessary because it shows to much detailed and the information is in the text.

6.     This experience can be an example for Latin America because of economy, distances, lack of specialists, etc…

Author Response

Dear reviewer,

Thank you for your constructive advice.

  • We have checked and corrected the information in part 3.3.2 (from line 306).
  • We have detailed the information on "Decision making": lines 261-270
  • 1 & 2 Thank you for this feedback!
  • 3. We have made references to other telemedicine teams in the text (line 73, 373-389)
  • 4. Finally, in the conclusion part there is a section with a focus on the results from the patient's perspective (line 440-450)
  • 5. We decided to leave the table in the text for reasons of clarity.
  • 6. We agree: and included the reference to Latin Amercia in line 449.

Thank you once more and best regards

Tabea Troschke

Reviewer 3 Report

This is a well-structured report for a meaningful project of developing, implementing, and managing a German-Polish telemedicine network in pediatric oncology and hematology. The needs for such service were well-justified. The procedures for the development of telemedicine networks were delineated in detail. I only have some minor comments as follows.

1.     More information should be provided for the target groups stated on page 4.

2.     Page 4, line 170, the frequency and duration of the meeting held by the working groups should be stated clearly in this report instead of stating “met at regular intervals’.

3.     Page 4, line 171, details in developing the teaching canon should also be provided. What were those needs identified?

4.     Page 5, lines 184-185, how many patients and their caregivers were recruited? How were they recruited for this project? More elaboration is needed.

5.     How were the curricula developed? Was it underpinned by any theoretical/conceptual framework/professional advice?

6.     Any barriers identified at the organization level? If any, how were they solved?

7.     In the results section, qualitative feedback “highly satisfied with the procedure, the content of the meetings as well as 251 the opportunity for them to witness the expert panel’s work, and to get the opportunity to ask questions” were reported, quotes could be provided in the results section to allow the readers have a more concrete understanding.

Author Response

Dear reviewer,

Thank you for your constructive advice and questions.

  1. we added more information regarding the frenquency (line 177), since the duration varied greatly, it was impossible to make a specific statement here.
  2. & 5. we added details on curriculum development (line 177-184)
  3. More information on the recruitment process has been added to the text (line 202-206)
  4. We refer to the identified barriers at the organizational level in line 97-101, 202 (and following) and 392(and following)) and table 6 (overview)
  5. We added facts from the family interviews. (line 280-285)
    1. Regarding more information for the target group from our point of view the most important facts are already given: what information would you need exactly?

Thank you once more and best regards

Tabea Troschke